# Monocytic TLR4 expression and activation in schizophrenia: A systematic review and meta-analysis

Melika Jameie[1,2]*, Sanaz Bordbar[3,4], Reza Samiee[2,5], Mobina Amanollahi[5,6], Mehdi Azizmohammad Looha[7], Mir Sajjad Aleyasin[3], Mohammad Reza Abdol Homayuni[5,8], Mehrdad Mozafar[5], Seyed Behnamedin Jameie[1]*, Shahin Akhondzadeh[9]*

1 Neuroscience Research Center, Iran University of Medical Sciences, Tehran, Iran, 2 Iranian Center of Neurological Research, Neuroscience Institute, Tehran University of Medical Sciences, Tehran, Iran, 3 Students' Scientific Research Center, Tehran University of Medical Sciences, Tehran, Iran, 4 Interdisciplinary Neuroscience Research Program, Tehran University of Medical Sciences, Tehran, Iran, 5 School of Medicine, Tehran University of Medical Sciences, Tehran, Iran, 6 Translational Ophthalmology Research Center, Farabi Eye Hospital, Tehran University of Medical Sciences, Tehran, Iran, 7 Basic and Molecular Epidemiology of Gastrointestinal Disorders Research Center, Research Institute for Gastroenterology and Liver Diseases, Shahid Beheshti University of Medical Sciences, Tehran, Iran, 8 NCweb association, Students' Scientific Research Center, Tehran University of Medical Sciences, Tehran, Iran, 9 Psychiatric Research Center, Roozbeh Hospital, Tehran University of Medical Sciences, Tehran, Iran

☯ Sanaz Bordbar and Reza Samiee contributed equally to this work and share the second authorship.
* jameie.sb@iums.ac.ir, behjame@gmail.com (SB); sakhond@yahoo.com (SA)

## Abstract

### Background

The role of toll-like receptor 4 (TLR4) in schizophrenia remains unclear, with studies reporting conflicting results on its expression and activation in persons with schizophrenia (PwSCZ). This systematic review/meta-analysis compared basal monocytic TLR4 expression, as well as its activation pattern between PwSCZ and healthy controls (HCs).

### Methods

This study was registered with PROSPERO (CRD42021273858) and adhered to the PRISMA guidelines. A systematic search was conducted through MEDLINE (via PubMed), Web of Science, and Scopus from inception to December 12, 2023. Quantitative syntheses were conducted for (a) basal monocytic TLR4 density, (b) basal percentage of TLR4⁺ monocytes, and (c) basal TLR4 gene expression. Effect sizes were computed using Hedges' g for mean differences. Random-effect models with restricted maximum-likelihood estimation were used, and subgrouping was conducted based on antipsychotic status. The studies' risk of bias was assessed using the Joanna Briggs Institute (JBI) tool.

### Results

Eleven studies (473 PwSCZ, 416 HCs) were included. Pooled analysis revealed a nonsignificant trend toward increased basal monocytic TLR4 density in PwSCZ (Hedges' g = 0.317 [95% CI: −0.060, 0.694], $\tau2 = 0.127$, $I^2 = 68.91\%$). The difference

**Data availability statement:** The data underlying the findings is uploaded as supporting information.

**Funding:** The author(s) received no specific funding for this work.

**Competing interests:** The authors have declared that no competing interests exist.

**Abbreviations:** AP, Antipsychotic; BMI, Body mass index; CNS, Central nervous system; DAMP, Damage-associated molecular pattern; HC, Healthy control; IL, Interleukin; JBI, Joanna Briggs Institute; LPS, Lipopolysaccharide; MFI, Mean fluorescent intensity; mRNA, Messenger ribonucleic acid; REML, Restricted maximum-likelihood; NF-κβ, Nuclear factor-kappa B; PAMP, Pathogen-associated molecular pattern; PANSS, Positive and Negative Syndrome Scale; PBMC, Peripheral blood mononuclear cell; PICO, Patient, Intervention, Comparison, and Outcome; polyI:C, Polyinosinic: polycytidylic acid sodium salt; PRISMA, Preferred Reporting Items for Systematic Reviews and Meta-Analyses; PROSPERO, International prospective register of systematic reviews; PwSCZ, Persons with schizophrenia; RoB, Risk of bias; RT-PCR, Reverse transcription polymerase chain reaction; SANS, Scale for Assessment of Negative Symptoms; SAPS, Scale for Assessment of Positive Symptoms; TLR, Toll-like receptor; TNF, Tumor necrosis factor.

became significant after sensitivity analysis and excluding one study (Hedges' g = 0.469 [0.195,0.742], p = 0.001). No significant difference was found between the groups in terms of TLR4+ monocytes percentage (Hedges' g = 0.235 [−0.245, 0.715], τ2 = 0.31, I² = 87.30%) or TLR4 gene expression (Hedges' g = 0.179 [−0.502, 0.861], τ2 = 0.29, I² = 79.04%). According to qualitative synthesis, TLR4 stimulation resulted in reduced monocytic activation in PwSCZ compared to HCs.

## Conclusions

This study suggested a trend toward an increased basal monocytic TLR4 density in PwSCZ, with no difference in the basal percentage of TLR4+ monocytes or TLR4 gene expression. However, the limited available data underscores the need for future studies.

## 1. Introduction

Schizophrenia is a debilitating mental disorder with a multifaceted pathophysiology [1,2]. It affects 24 million people worldwide, with a global age-standardized prevalence and incidence of 287.4 and 16.31 per 100,000 persons, respectively [3]. Although the pathogenesis of schizophrenia is unknown, it likely results from an intricate interplay between genetic and external factors (i.e., infections, inflammation, and altered immune system activation) [4]. Prior evidence established a link between immune system abnormalities and various central nervous system (CNS) disorders [5–8], paving the way for novel therapeutic options [9–12]. Subsequently, there has been considerable emphasis on the importance of disruptions in innate immunity [8], particularly Toll-like receptors (TLRs), in the pathogenesis of schizophrenia [13,14].

TLRs are expressed in various cells, including monocytes, macrophages, and glial cells [15], playing vital roles in the innate immune system [16]. Extracellular TLRs interact with extracellular bacterial components, and intracellular TLRs engage with the nucleic acids of infiltrating bacteria and viruses [17]. TLRs play a key role in recognizing damage-associated molecular patterns (DAMPs) and pathogen-associated molecular patterns (PAMPs) [18]. Upon activation, TLRs initiate a complex intracellular signaling cascade, leading to the production of proinflammatory cytokines [19]. TLR4 is an extracellular transmembrane receptor with high expression in monocytes [18]. It recognizes host-derived DAMPs, as well as various PAMPs, including the lipopolysaccharide (LPS) of gram-negative bacteria [18]. Activation of TLR4 by its ligands causes the translocation of nuclear factor-kappa B (NF-κβ) and the production of proinflammatory cytokines, including interleukin (IL)-6, tumor necrosis factor (TNF)-α and IL-1β [20].

Recent studies have suggested a diminished monocytic TLR4 response to LPS stimulation in individuals with schizophrenia, with increased TLR4 expression as a compensatory mechanism for this attenuated monocytic activation [21,22]. Polymorphisms in the TLR4 gene have also been linked to increased susceptibility to schizophrenia [23]. Postmortem studies have further revealed altered TLR4 expression and signaling in the brains of individuals with schizophrenia [14]. Altered TLR4 expression and function may contribute to schizophrenia by affecting the innate immune system activation [14], neuroinflammation [24], increased susceptibility to infection [25,26], and structural brain changes [27]. Higher monocytic TLR4 expression in people with schizophrenia [21,22,28,29] may contribute to neuroinflammation [14,24,30]. Subsequently, the altered inflammatory response may contribute to the adverse effects of schizophrenia on the brain, including cognitive function [9,14]. At the same time,

while TLR4 might be overexpressed in persons with schizophrenia, its stimulation might induce a weaker increase in TLR4 expression and reduced proinflammatory cytokine production compared to healthy controls (HCs) [21,22,30,31]. This suggests that individuals with schizophrenia may be less able to react appropriately to infections [25,26], which could result in chronic inflammatory states that worsen symptoms of schizophrenia [30].

Despite recent investigations [21,22,28–36], a comprehensive understanding of how TLR4 contributes to schizophrenia has not been achieved. Additionally, studies on TLR4 expression in persons with schizophrenia have generated controversial findings, reporting increased, decreased, or unchanged TLR4 expression. Several factors may have contributed to these discrepancies, including variations in antipsychotic (AP) medications and participant characteristics. Recognizing these issues, along with the potential role of TLR4 as a novel therapeutic target [13,20,37], we conducted a systematic review and meta-analysis of the literature on the basal expression and activation of TLR4 in peripheral blood mononuclear cells (PBMCs) in individuals with schizophrenia. The aim was to summarize existing literature, identify knowledge gaps, and focus on the dysregulated immune response, particularly involving TLR4, which may significantly contribute to the pathophysiology of schizophrenia. This review may inspire future research into developing innovative therapeutic strategies. Such strategies may involve modulating TLR4 activity or targeting related downstream pathways as adjunctive treatments alongside standard APs.

## 2. Methods

### 2.1. Objectives and review questions

We conducted a systematic review and meta-analysis to investigate the basal expression and activation pattern of TLR4 in PBMCs, including monocytes, among individuals with schizophrenia compared to HCs. We used the "Patient, Intervention, Comparison, and Outcome" (PICO) model to formulate our research question [38]:

- *Patient*: Persons with schizophrenia

- Intervention/exposure: not applicable

- *Comparison*: HCs

- *Outcome(s)*:

  1. TLR4 *basal expression* patterns in PBMCs (in unstimulated cells)

  2. TLR4 *activation* patterns in PBMCs (in stimulated cells)

### 2.2. Study design, information sources, and search strategy

This study is reported according to the *Preferred Reporting Items for Systematic Reviews and Meta-Analyses* (PRISMA) framework [39] (Supporting Information 1 and Supporting Information 2; Tables S1 and S2). A systematic search was conducted in MEDLINE (via PubMed), Scopus, and Web of Science from inception to December 12, 2023. No filters were applied during the search process. Two PICO elements were used to convert our research question to a search strategy [38], including "patient"-related subject headings (MeSH Terms) and text word terms ("schizophrenia", "schizophrenic", "schizophrenias", "psychosis", and "psychotic") and "outcome"-related subject headings and text word terms ("toll-like", "toll-like receptor", "toll-like receptor 4", and "TLR"). The detailed search strategy is presented in Supporting Information 2; Tables S3-S5. We also searched for references in reviews, editorials, letters,

conference papers, and references cited within the included studies through a manual search to find any initially unretrieved relevant studies. The use of previously published data in this study resulted in an exemption from ethical approval by the local institutional ethics committee. The study protocol was registered with International Prospective Register of Systematic Reviews (PROSPERO: CRD42021273858).

### 2.3.  Eligibility criteria

All studies employing blood samples to assess TLR4 basal expression or TLR4 activation in persons with schizophrenia were considered eligible for inclusion. *In vitro* studies, animal studies, postmortem studies, case reports, case series, commentaries, conference papers, non-English studies, and review articles were excluded. Additionally, articles not aligning with our research question, including investigations into TLR4-related outcomes differing from our specific focus (e.g., TLR4 gene polymorphisms) were excluded.

### 2.4.  Selection process

Records from the databases were initially imported into EndNote software (version X9, Clarivate Plc). After removing duplicates, two independent researchers (R.S. and S.B.) screened all the records using titles and abstracts. Subsequently, the full texts of potentially eligible records were further assessed (M.J., S.B., and R.S.). The records that did not align with the predefined eligibility criteria were excluded. In cases of discrepancies, conflicts were resolved by reaching a consensus.

### 2.5.  Data collection

The following data were extracted: study-related variables (authors, study location, publication year, sample size, and study design and setting), individuals' demographic and clinical characteristics (age, sex, schizophrenia symptom scores, including the Positive and Negative Syndrome Scale [PANSS], Scale for Assessment of Positive Symptoms [SAPS], and Scale for Assessment of Negative Symptoms [SANS], disease onset age, disease duration, and medications for schizophrenia), and TLR4-related variables (method used to evaluate the expression/activation of TLR4, and TLR4 expression levels before and after stimulation). Three authors (M.J., R.S., and S.B.) independently extracted the data, and conflicts were resolved via consensus between two authors (M.A. and M.J.). Notably, all data for the meta-analyses were extracted from the main text of the original manuscripts. When numerical values were not reported in the text, they were obtained from figures using WebPlotDigitizer. As a result, these extracted values may differ slightly from the original numbers reported by the authors. The data supporting the findings is provided in Supporting Information 3.

### 2.6.  Risk of bias assessment

The Joanna Briggs Institute (JBI) Critical Appraisal tool was utilized by three independent researchers (S.B., M.J., and MR.AH.) to assess the study's risk of bias (RoB) [40], and disagreements were resolved through consensus. This tool assesses comparability between cases and controls in terms of parameters other than the presence of the disease, including age and sex (questions one and two), consistency in the case and control identification criteria (question three), validity and uniformity of exposure measurement and outcome assessment for both groups (questions four, five, and eight), identification and adjustment for confounding factors (questions six and seven), significance of the exposure period (question nine), and utilizing proper statistical analysis (question ten).

## 2.7. Data synthesis

For qualitative assessment, the included studies were reviewed in two sections:

1. Studies addressing *basal* monocytic TLR4 expression (in unstimulated cells). There were three outcomes of interest:

   a) The surface TLR4 density on the peripheral blood monocyte membrane (assessed by the mean fluorescent intensity [MFI]).

   b) Percentage of peripheral blood monocytes expressing TLR4 (TLR4+ monocytes).

   c) Gene/messenger ribonucleic acid (mRNA) expression of TLR4 in PBMCs.

2. Studies addressing *stimulated* monocytic TLR4 expression after activation with its ligand(s).

Quantitative synthesis was performed by comparing (a) *basal* surface monocytic TLR4 density (MFI), (b) *basal* percentage of TLR4-expressing (TLR4+) monocytes, and (c) *basal* TLR4 gene expression between persons with schizophrenia and HCs. Considering the suggested impact of APs on immune parameters, including the TLR4 signaling pathway [41], we also considered the AP status of individuals when conducting the meta-analyses.

## 2.8. Statistical analysis

For quantitative synthesis (meta-analysis), statistical analyses were performed using the 'meta' package version 5.2–0 in R software version 4.2.1. We calculated effect sizes using Hedges' g to evaluate mean differences. This involved employing a random-effects model with restricted maximum-likelihood (REML) estimation, and subgrouping was performed based on AP status. Sensitivity analyses involving iterative removal of one study at a time and subsequent reanalysis (leave-one-out analysis) were also conducted to assess the potential impact of individual studies on the overall pooled results. The between-study heterogeneity was evaluated by Cochran's Q statistic, $\tau2$, and the $I^2$ index [42,43]. Furthermore, the Galbraith plot was used to evaluate heterogeneity among the effect sizes and the robustness of studies concerning this heterogeneity. Publication bias was assessed with funnel plots and Begg's test for small-study effects [44]. A two-tailed P value < 0.05 was considered to indicate statistical significance.

## 3. Results

### 3.1. Literature search and study selection

An overview of the inclusion process is depicted in the PRISMA flowchart (Fig 1). Our search resulted in 622 records. After 170 duplicate deletions, 452 articles were screened by title and abstract, of which 327 articles did not meet the eligibility criteria. Hence, 125 articles remained for full-text screening, 114 of which were excluded for the following reasons: not evaluating TLR4-related outcomes (n = 24), evaluating outcomes in populations other than people with schizophrenia (n = 5), being non-original articles (n = 64), animal studies (n = 9), in vitro studies (n = 5), post-mortem studies (n = 4), and non-English studies (n = 2). Additionally, one study was excluded for evaluating TLR4 gene expression in peripheral blood leukocytes rather than specifically in PBMCs [45]. Table S6 in Supporting Information 2 presents a numbered list of the 125 studies identified in the literature search that proceeded to full-text screening, along with the reason(s) for exclusion for each excluded study. No additional studies were identified through manual searches of reference lists of the included studies or review articles. Finally, 11 articles remained for qualitative synthesis. Quantitative syntheses were conducted in three sections. The first section included four studies comparing basal TLR density on the

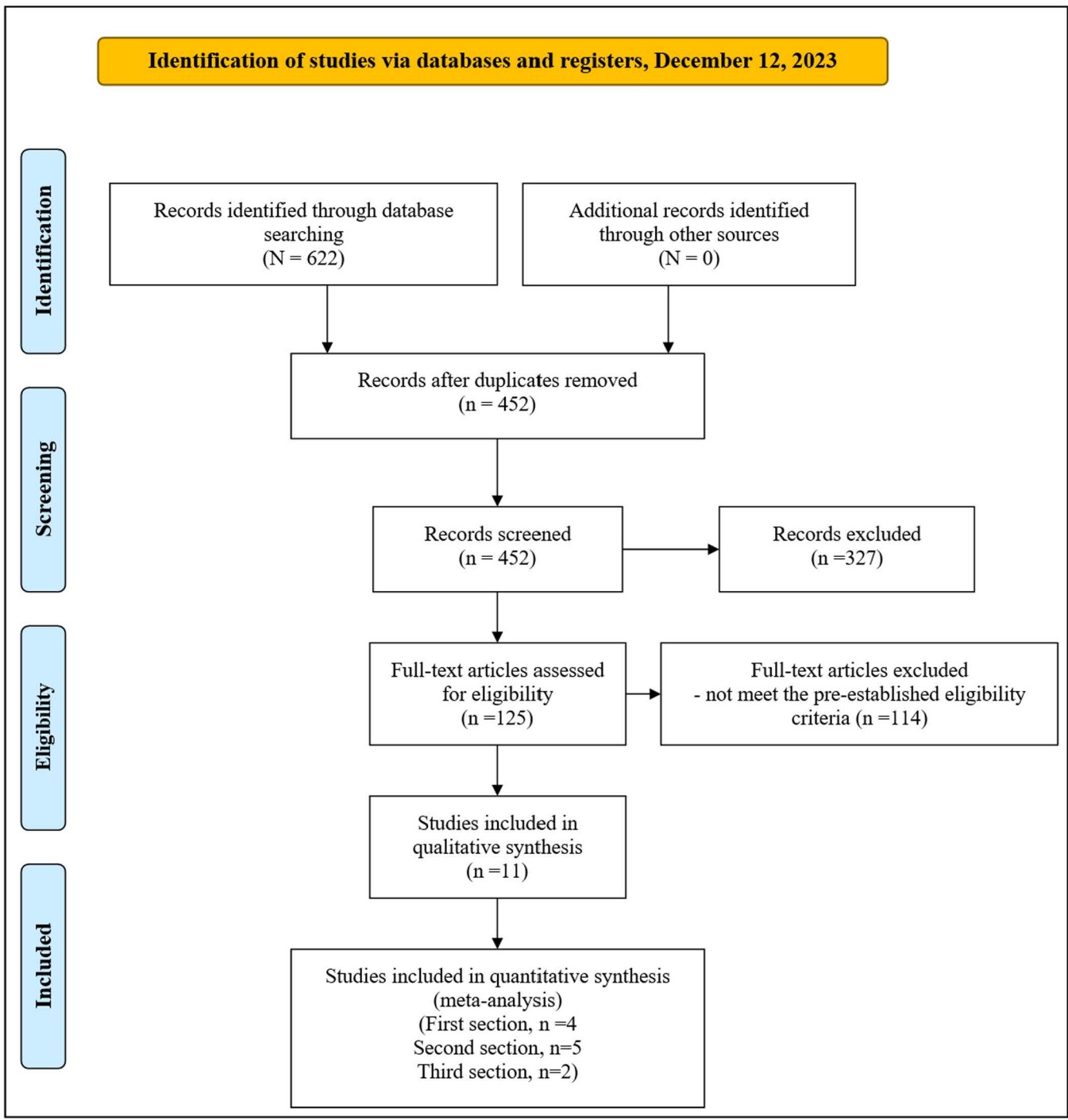

**Fig 1. The PRISMA flow diagram.** From: Page MJ, McKenzie JE, Bossuyt PM, Boutron I, Hoffmann TC, Mulrow CD, et al. The PRISMA 2020 statement: an updated guideline for reporting systematic reviews. BMJ 2021;372:n71. https://doi.org/10.1136/bmj.n71.

monocyte membrane (MFI) between persons with schizophrenia and HCs [21,22,28,31]. The second section included five studies pooling data on the basal percentage of TLR4+ monocytes [28,30,31,33,36], and the third section included combined results from two studies comparing basal TLR4 gene expression between persons with schizophrenia and HCs [29,34].

## 3.2. Study characteristics

Eleven studies (889 individuals; 473 persons with schizophrenia and 416 HCs) dealing with *basal* TLR4 expression were assessed [21,22,28–36] (Table 1). Four of the eleven studies (439 individuals; 239 persons with schizophrenia and 200 HCs) also investigated *stimulated* TLR4 expression after activation by its ligand(s) [22,30,31,33] (Table 2).

Most of the individuals in the studied population were men (persons with schizophrenia: 309/473 (65.3%), HCs: 245/416 (58.9%)). Using Cochrane's formulation [46], the combined mean ± SD ages of persons with schizophrenia and HCs were 37.90 ± 12.34 and 36.12 ± 12.06, respectively. Some of the studies assessed their outcome in first-episode persons with schizophrenia [21,28,31], while others included individuals with chronic schizophrenia (those with either stable schizophrenia or present-state psychosis) [30,32–36]. The mean duration of schizophrenia varied between 7.6 months [21] and 25.5 years [33]. The PANSS or SAPS/SANS scores were reported in all studies except for three [32,34,36]. Six studies included participants who were taking APs at the time of the study [30,32–36]. However, some other studies included drug-naïve participants [21] or participants free of APs at the time of the study [22]. Two studies assessed both conditions [28,29], and one study included a combination of drug-naïve persons with schizophrenia and those who received APs for a short duration (<10 days) [31]. Flow cytometry was used in seven studies to measure the TLR4 density on monocyte membrane [21,22,28,31] or the percentage of TLR4$^+$ monocytes [28,30,31,33,36]. Additionally, four studies used reverse transcription polymerase chain reaction (RT–PCR) to assess TLR4 gene/mRNA expression [29,34,35] or determine the percentage of individuals with monocytes exhibiting TLR4 mRNA expression [32].

## 3.3. Risk of bias (RoB) in studies

Table S7 in Supporting Information 2 illustrates the RoB in the included studies. To evaluate the comparability between the groups (q one and two), we considered key characteristics identified in the literature that could considerably impact TLR4 expression, including age [47], sex [48], body mass index (BMI) [49], and smoking status [50]. Groups were deemed comparable when there were no significant differences in at least three of these characteristics. Five studies utilized individual matching [21,22,28,35,36]. Two studies matched participants for age, sex, and education [21,28], one for age, sex, and ethnicity [22], one for metabolic parameters [35], and one for age [36]. In seven studies [21,22,28,31,33,34,36], the groups were comparable. However, in two studies [32,35], they were not comparable, and in two studies [29,30], information on BMI and smoking was not reported. All studies used the same criteria for the identification of cases and controls (q three). Exposure was measured validly and reliably in all studies, except one (q four) [32]. Exposure was measured in the same way for cases and controls (q five) and was long enough to be meaningful (q nine) in all the studies. Confounding factors were identified and adjusted properly in all but two studies (q six and seven) [22,32]. Standard outcome assessment was implied in all studies except one (q eight) [32]. There were more appropriate alternate statistical methods that could have been used in three studies (q ten) [22,28,32].

## 3.4. Basal surface monocytic TLR4 density: individual findings and quantitative synthesis

Four studies (289 individuals; 150 persons with schizophrenia and 139 HCs) compared basal TLR density on the monocyte membrane between persons with schizophrenia and HCs using MFI [21,22,28,31]. Three studies assessed the outcome in AP-free individuals [21,22,31], and one assessed the outcome in both drug-naïve participants and the same participants after

**Table 1. Studies addressing *basal* TLR4 expression in persons with schizophrenia and healthy controls.**

| Author (year) | Country | Design | Setting | Sample size | Sex (M, %) | Age mean (SD) | Symptoms severity (PANSS) mean (SD) | onset age, y/o mean (SD) | disease duration mean (SD) | Medication (for SCZ) | Sample | Method | Outcome measure(s) | Result mean ± SD, median [IQR], n (%) | P* |
|---|---|---|---|---|---|---|---|---|---|---|---|---|---|---|---|
| **Chang et al. (2011)** [32] | Taiwan | Case-control | SCZ with present psychosis, inpatient | HC: 22 / SCZ: 46 | 22 (100) / 46 (100) | 33 (5.88) / 40 (10.42) | NA | 30 (9.81) | 10 y (6.30) | taking at least one AP | PB – Monocyte | RT–PCR | persons with monocytic TLR4 mRNA expression (%) | HC: 21/22 (95.5%)† ; SCZ: 46/46 (100%)† | NS P: N/A |
| **Müller et al. (2012)** [22] | Germany | Case-control, (age, sex, ethnicity-matched) | acute SCZ, inpatient | 31 / 31 | 18 (58) / 18 (58) | 33.7 (16.1) / 36.7 (13.6) | T: 92.1 (20.3) | 31.9 (12.9) | NA | free of APs for ≥ 4 w | PB – Monocyte | Flow cytometry | Surface monocytic TLR4 density(MFI) | HC: 26.8 ± 8.4‡ ; SCZ: 38.1 ± 23.6‡ | ↑ P ≤ 0.001 |
| **Kéri et al. (2016)** [28] ‡‡ | Hungary | Case-control, (age, sex, education-matched) | first episode patients | 30 / 35 | 21 (70) / 25 (71) | 26.4 (5.5) / 26.7 (7.0) | P: 19.1 (7.8) N: 13.7 (6.7) G: 57.1 (16.4) | NA | 8.0 m (3.2)§ | • drug-naïve (unmedicated)‖ • risperidone/ olanzapine (8w, medicated)‖ | PB – Monocyte | Flow cytometry | TLR4+ monocytes‖ (%) | *unmedicated:* * HC: 8.07 (95%CI: 6.59, 9.62( * SCZ: olanzapine: 12.60 (10.81, 14.17), risperidone: 15.23 (12.89, 17.74) ; *medicated:* * HC: 8.29 (7.18, 9.40) * SCZ: olanzapine: 9.31 (8.09, 10.52), risperidone: 8.89 (7.04, 10.75) | unmed: ↑ P < 0.01 ; med: NS P > 0.5 |
| | | | | | | | | | | | | | Surface monocytic TLR4 density (MFI)‖ | *Unmedicated:* * HC: 1183.30 (1056.45, 1315.03) *SCZ: olanzapine 1465.90 (1317.09, 1609.83), risperidone: 1677.73 (1465.51, 1899.72) ; *medicated:* * HC: 1179.63 (1080.83, 1278.43) * SCZ: olanzapine 1193.90 (1073.13, 1300.02), risperidone:1326.47 (1160.59, 1499.69) | unmed: ↑ P < 0.05 ; med: NS P > 0.5 |
| **Kéri et al. (2017)** [21] ‡‡ | Hungary | Case-control, (age, sex, education-matched) | first episode patients | 42 / 42 | 29 (69) / 29 (69) | 26.2 (5.9) / 26.1 (6.8) | P: 19.4 (7.9) N: 13.5 (7.0) G: 56.3 (17.1) | NA | 7.6 m (4.0)¶ | drug-naïve | PB – Monocyte | Flow cytometry | Surface monocytic TLR4 density(MFI) | HC: 1197.64 (1099.21, 1300.79) ; SCZ: 1599.21 (1499.21, 1700.79) | ↑ P < 0.001 |
| **Chen et al. (2019)** [31] | China | Case-control | first episode patients | 36 / 42 | 21 (58) / 18 (43) | 26.47 (4.40) / 25.21 (6.20) | T: 80.43 (12.51) P: 22.18 (5.97) N: 18.78 (6.53) G: 39.47 (7.37) | NA | 16.1 m (5.4) | • never been exposed to APs: 5/42 • received < 10 d of APs: 37/42 | PB – Monocyte | Flow cytometry | TLR4+ monocytes (%)†† ; Surface monocytic TLR4 density (MFI)†† | HC: 82.51 ± 8.64‡‡ ; SCZ: 80.02 ± 10.05‡‡ ; HC: 18.38 ± 3.84‡‡ ; SCZ: 17.19 ± 3.97‡‡ | NS P:0.238 ; NS P:0.183 |

*(Continued)*

**Table 1.** (Continued)

| Author (year) | Country | Design | Setting | Sample size | Sex (M, %) | Age mean (SD) | Symptoms severity (PANSS) mean (SD) | onset age, y/o mean (SD) | disease duration mean (SD) | Medication (for SCZ) | Sample | Method | Outcome measure(s) | Result mean ± SD, median [IQR], n (%) | P* |
|---|---|---|---|---|---|---|---|---|---|---|---|---|---|---|---|
| **H. Li et al. (2022) [30]** | China | Case-control | stable chronic SCZ (inpatients/ outpatients) | 59 | 28 (47) | 43.54 (11.38) | T: 50.64 (10.67) P: 11.27 (4.05) N: 15.07 (5.65) G: 24.32 (3.64) | 24.1 (5.92) | 22.91 m (12.19) | a steady dose of APs for ≥ 6 m | PB – Monocyte | Flow cytometry | TLR4+ mono-cytes (%) | HC: 52.98±0.26[‡‡] | NS P:0.284 |
| | | | | 44 | 29 (66) | 47.02 (10.7) | | | | | | | | SCZ: 57.76±26.48[‡‡] | |
| **N. Li et al. (2022) [33]** | China | Case-control | chronic SCZ (inpatients/ outpatients) | HC:74 | 45 (96) | 48.15 (8.10) | TD: T: 67.64 (14.25) P: 16.57 (6.31) N: 19.67 (4.15) G: 31.39 (7.10) | TD: 23.4 (6.30) | TD: 25.50 y (8.33) | taking a steady dose of oral APs for ≥ 6 m[§§] | PB – Monocyte | Flow cytometry | TLR4+ mono-cytes (%) | HC: 51.97[‡‡] [39.01, 70.75] | TD: ↓ P:0.004 |
| | | | | TD: 61 | 45 (74) | 48.43 (9.21) | | | | | | | | SCZ, TD: 35.82[‡‡] [25.28, 56.82] | |
| | | | | NTD:61 | 35 (57) | 46.21 (8.94) | NTD: T: 65.52 (16.00) P: 15.52 (7.19) N: 18.87 (5.22) G: 31.13 (7.89) | NTD: 23.8 (5.76) | NTD: 23.55 y (9.82) | | | | | HC: 51.97[‡‡] [39.01, 70.75] | NTD: ↓ P:0.001 |
| | | | | | | | | | | | | | | SCZ, NTD: 34.64[‡‡] [24.87, 70.11] | |
| **Tsai et al. (2023) [36]** | Taiwan | Case-control (age-matched) | Clinically stable outpatients | 43 | 13 (30.2) | 36.8 (7.8) | NA | NA | NA | APs for at least 3 m | PB – Monocyte (classical)[∥∥] | Flow cytometry | TLR4+ mono-cytes (%) | SCZ: 2.1 ±3.4 HC: 0.1±0.2 | ↑ P < 0.025 |
| | | | | 33 | 18 (54.5) | 36.9 (7.1) | | | | | PB – Monocyte (intermedi-ate)[∥∥] | | | SCZ: 0.6 ±1.8 HC: 0.0±0.0 | NS P: 0.064 |
| **Balaji et al. (2019) [29]** | India | Case-control | drug-naïve individuals diagnosed with SCZ (outpatients) | 30 | 18 (60) | 27.73 (3.63) | SAPS: 27.97 (12.31) SANS: 32.00 (26.20) | 28.42 (7.71) | 41.71 m (62.03)[**] | • drug-naïve (unmedicated)[∥] • APs (3 m, med-icated)[∥] | PBMC | RT-PCR | TLR4 gene expression | HC, unmedicated: 0.75±0.41 | unmed: ↑ P:0.05 |
| | | | | | 31 | 18 (58) | 32.23 (7.85) | | | | | | | SCZ, unmedicated: 0.93±0.35 | |
| | | | | | | | | | | | | | SCZ, unmedicated: 0.93±0.35 | NS P:0.77 |
| | | | | | | | | | | | | | SCZ, medicated: 0.95±0.28 | |
| **Chase et al. (2019) [34]** | USA | Case-control | patients with SCZ with present state psychosis | 20 | 10 (50) | 38.15 (13.52) | – | 22.47 (8.70) | 15.84 y (14.31) | actively taking APs | PBMC | RT-PCR | TLR4 gene expression | HC: 2.38±0.53 | NS P:0.08 |
| | | | | 20 | 10 (50) | 37.85 (12.38) | | | | | | | | SCZ: 2.12±0.37 | |

*(Continued)*

**Table 1.** (Continued)

| Author (year) | Country | Design | Setting | Sample size | Sex (M, %) | Age mean (SD) | Symptoms severity (PANSS) mean (SD) | onset age, y/o mean (SD) | disease duration mean (SD) | Medication (for SCZ) | Sample | Method | Outcome measure(s) | Result mean ± SD, median [IQR], n (%) | P* |
|---|---|---|---|---|---|---|---|---|---|---|---|---|---|---|---|
| Kozłowska et al. (2019) [35] | Poland | Case-control (metabolic matched) §§§ | Chronic paranoid SCZ (outpatients) | 29 | 20 (69) | 37.9 (10.6) | T: 65.4 (14.6) P: 15.3 (5.1) N: 18.0 (4.5) G: 32.1 (7.3) | N/A | 16.1 y (10.8)††† | taking at least one AP | PBMC | RT-PCR | TLR4 mRNA expression | HC: 262.36 ± 290.1 | ↓ P < 0.001 |
| | | | | 27 | 18 (67) | 38.6 (9.3) | | | | | | | | SCZ: 18.90 ± 11.71 | |

*Abbreviations*: AP: antipsychotic; d: day; G: general; HC: healthy control; m: month; MFI: mean fluorescent intensity; mRNA: messenger RNA; N: negative; NA: not available; NS: not significant; NTD: without tardive dyskinesia; P: positive; PANSS: positive and negative syndrome scale; PB: peripheral blood; PBMC: peripheral blood mononuclear cell; PCR: polymerase chain reaction; RT–PCR: reverse transcription polymerase chain reaction; SANS: scale for assessment of negative symptoms; SAPS: scale for assessment of positive symptoms; SCZ: schizophrenia; SD: standard deviation; T: total; TD: tardive dyskinesia, TLR: toll-like receptor; y: year.

↑: Increased (in patients with schizophrenia compared to healthy controls); ↓: Decreased (in patients with schizophrenia compared to healthy controls).

*P values are obtained from the primary studies.

†In this study, the association between TLR4 mRNA expression and schizophrenic disorder was assessed using the chi-square test. Accordingly, 21/22 (95.5%) and 46/46 (100%) healthy individuals and patients with schizophrenia, respectively, expressed TLR4 mRNA expression, indicating no significant association (P > 0.05).

‡In this study, the authors investigated the cell-surface expression of TLR4 on CD14+ monocytes under unstimulated and simulated conditions (stimulated with both lipopolysaccharide (LPS) and polyI:C to mimic a bacterial or a viral infection, respectively). The numbers in this table are attributed to the unstimulated conditions.

§This study included patients with first-episode schizophrenia, none of whom received psychotropic medications before the assessment. However, the mean (SD) duration of psychosis with no treatment was 8.0 months (3.2).

∥In this study, the authors assessed the outcomes in two settings, including drug-naïve (unmedicated) patients with schizophrenia and the same patients after receiving antipsychotic treatment with risperidone or olanzapine for eight weeks. Two TLR4-related outcome measures were assessed in this study, namely, the percentage of monocytes expressing TLR4 and the TLR density on the cell membrane (assessed by MFI). The authors found a greater value in patients than in controls at baseline (p < 0.01) but not during follow-up (after treatment) (p > 0.5). During the follow-up period, the authors reported a significant decrease in the TLR4+ monocyte ratio in patients with schizophrenia (p < 0.001), whereas the control values remained similar (p > 0.5). Consistent results were obtained for the TLR density on the cell membrane.

¶This study included patients with first-episode schizophrenia, none of whom received psychotropic medications before the assessment. However, the mean (SD) duration of untreated psychosis was 7.6 months (4.0).

††In this study, two TLR4-related outcome measures were evaluated, namely, the percentage of monocytes expressing TLR4 and the TLR density on the cell membrane (assessed by MFI).

‡‡The authors investigated their outcomes of interest under unstimulated and simulated conditions (stimulated with LPS). The numbers in this table are attributed to the unstimulated conditions.

§§To limit the effect of antipsychotics on the TLR4 signaling pathway, the authors tried to ensure consistency in the dosage and ratio of the number of patients on first- versus second-generation antipsychotics among the TD and NTD groups at the beginning of the experiment.

∥∥In this study, the authors assessed the outcomes in two settings, including drug-naïve (unmedicated) patients with schizophrenia and the same patients after receiving antipsychotic treatment for 3 months. In the first setting, the authors assessed the differences in the gene expression of TLR4 between patients and healthy controls, while in the second setting, they evaluated the gene expression profile of TLR4 in patients before and after treatment.

¶¶This study included drug-naïve patients, none of whom received psychotropic medications before the assessment. However, the mean (SD) duration of untreated illness was 41.71 months (62.03).

†††The reported number is attributed to the duration of "treatment", and the total years of disease duration (considering the duration of untreated disease) is not specifically reported.

‡‡‡As the study results were shown in the figures (not tables), the numbers were extracted from the figures using https://automeris.io/WebPlotDigitizer/. Therefore, the extracted numbers might be slightly different from the exact numbers found by the authors.

*Note*: For MFI in the Keri 2016 and Keri 2017 studies, log¹⁰ was reported.

§§§Patients and controls were matched for metabolic parameters, including anthropometric indices, laboratory parameters, and body composition.

∥∥∥Classical monocytes: CD14+ and CD16- and intermediate monocytes: CD14+ and CD16+

**Table 2. Studies addressing *stimulated* TLR4 expression in persons with schizophrenia and healthy controls.**

| Author (year) | Country | Design | Setting | Sample size | Sex (M, %) | Age mean (SD) | Symptoms severity (PANSS) mean (SD) | onset age, y/o mean (SD) | disease duration mean (SD) | Medication (for SCZ) | Sample | Method | Stimula-tion dosage duration | Outcome measure(s) | Result mean±SD, median [IQR] | P* |
|---|---|---|---|---|---|---|---|---|---|---|---|---|---|---|---|---|
| **Müller et al. (2012) [22]** | Germany | Case-control, (age, sex, ethnicity match) | acute SCZ, inpatient | HC: 31 SCZ: 31 | 18 (58) 18 (58) | 33.7 (16.1) 36.7 (13.6) | T: 92.1 (20.3) | 31.9 (12.9) | NA | All were free of APs for ≥ 4 w | PB – Monocyte | Flow cytometry | LPS (1 µg/ml) 4 h | Surface monocytic TLR4 density after LPS stim, (MFI) | HC: 73.7±55.7 SCZ: 87.5±44.42 | NS P:0.092 |
| | | | | | | | | | | | | | | *increase* in mono-cytic TLR4 density after LPS stim[†] | HC: 2.10±1.03 SCZ: 1.47±0.76 | ↓ ‡ P:0.004 |
| | | | | | | | | | | | | | polyI: C (50 µg/ml) 4 h | Surface monocytic TLR4 density after polyI: C stim, (MFI) | HC: 74.0±56.1 SCZ: 86.3±40.5 | NS P:0.061 |
| | | | | | | | | | | | | | | *increase* in mono-cytic TLR4 density after polyI: C stim[†] | HC: 2.12±0.85 SCZ: 1.45±0.69 | ↓ ‡ P ≤ 0.001 |
| **Chen et al. (2019) [31]** | China | Case-control | first episode patients | 36 42 | 21 (58) 18 (43) | 26.47 (4.40) 25.21 (6.20) | T: 80.43 (12.51) P: 22.18 (5.97) N: 18.78 (6.53) G: 39.47 (7.37) | NA | 16.1 m (5.4) | •never exposed to APs: 5/42 •received <10 d of APs: 37/42 | PB – Monocyte | Flow cytometry | LPS (100 ng/ml) 5 h | TLR4+ monocytes after stim (%) | HC: 81.34±12.45 SCZ:76.33±12.56 | ↓ P:0.047 |
| | | | | | | | | | | | | | | Reduce rate of TLR4+ monocytes [I] (%) | HC: 18±12 SCZ: 23 ±12 | NS P:0.053 |
| | | | | | | | | | | | | | | Surface monocytic TLR4 density after stim (MFI) | HC: 36.14±4.46 SCZ: 33.94±3.89 | ↓ P:0.031 |
| | | | | | | | | | | | | | | Reduce rate of MFI [I] (%) | HC: 39±8 SCZ:40±9 | NS P:0.492 |
| **H. Li et al. (2022) [30]** | China | Case-control | stable chronic SCZ (inpatients/ outpatients) | 59 44 | 28 (47) 29 (66) | 43.54 (11.38) 47.02 (10.7) | T: 50.64 (10.67) P: 11.27 (4.05) N: 15.07 (5.65) G: 24.32 (3.64) | 24.1 (5.92) | 22.91 m (12.19) | taking a steady dose of APs for ≥6 m | PB – Monocyte | Flow cytometry | LPS (100 ng/ml) 5 h | TLR4+ monocytes after stim (%) | HC:87.13±11.77 SCZ:66.93±22.04 | ↓ P < 0.001 |
| **N. Li et al. (2022) [33]** | China | Case-control | chronic SCZ (inpatients) (outpatients) | HC:74 TD: 61 | 45 (96) 45 (74) | 48.15 (8.10) 48.43 (9.21) | TD+: T: 67.64 (14.25) P: 16.57 (6.31) N: 19.67 (4.15) G: 31.39 (7.10) | TD+: 23.4 (6.30) | TD+: 25.50 y (8.33) | taking a steady dose of oral APs for ≥6 m[§§] | PB – Monocyte | Flow cytometry | LPS (100 ng/ml) 5 h | Reduce rate of TLR4+ monocytes (%)[¶] | HC:0.56 [0.24–1.02] SCZ, TD: 0.74 [0.16–1.45] | NS P:0.687 |
| | | | | NTD:61 | 35 (57) | 46.21 (8.94) | TD -: T: 65.52 (16.00) P: 15.52 (7.19) N: 18.87 (5.22) G: 31.13 (7.89) | TD-: 23.8 (5.76) | TD-: 23.55 y (9.82) | | | | | | SCZ, NTD: 0.72 [-0.12–1.43] | |

*Abbreviations:* AP: antipsychotic; d: day; G: general; HC: healthy control; LPS: lipopolysaccharide; m: month; MFI: mean fluorescent intensity; N: negative; NA: not available; NS: not significant; NTD: without tardive dyskinesia; P: positive; PANSS: positive and negative syndrome scale; PB: peripheral blood; PBMC: peripheral blood mononuclear cells; polyI:C: polyinosinic: polycytidylic acid sodium salt; SCZ: schizophrenia; SD: standard deviation; stim: stimulation; T: total; TD: tardive dyskinesia; TLR: toll-like receptor; y: year.

†: Increased (in patients with schizophrenia compared to healthy controls); ↓: Decreased (in patients with schizophrenia compared to healthy controls).

*P values are obtained from the primary studies.

†Increase in TLR4 expression was assessed using the following formula: unstim/stim, indicating the difference between the stimulated and unstimulated expression of TLR4.

‡This result indicates a significantly less increase in TLR4 expression in patients with schizophrenia than in controls.

ℙThe reduction rate after stimulation with LPS was assessed using the following formula: (unstimulated − stimulated)/unstimulated.

¶The relationship between the TLR4 pathway before and after stimulation was assessed using the following formula: (stimulated-unstimulated)/unstimulated

receiving AP treatment [28]. In AP-free settings [21,22,28,31], all studies, except one [31], reported increased monocytic TLR4 expression in persons with schizophrenia compared to HCs [21,22,28]. After eight weeks of AP treatment, Keri *et al.* found no significant difference in monocytic TLR4 density between individuals with schizophrenia and HCs [28].

Fig 2 displays forest plots illustrating the effect size (Hedges' g) of outcomes between individuals with schizophrenia and healthy controls. Subgroup meta-analysis revealed positive effect sizes in both the AP-free (combined effect size: 0.345, 95% CI: -0.137 to 0.827) and AP-medicated (combined effect size: 0.218, 95% CI: -0.265 to 0.701) subgroups. The overall pooled effect size of 0.317 (95% CI: -0.060 to 0.694) indicated a nonsignificant trend toward greater TLR4 density in individuals with schizophrenia. The test of group differences showed no significant difference ($Q_b)1(= 0.13$, $P = 0.715$), implying no significant difference between the AP-free and AP-medicated subgroups among individuals with schizophrenia (Fig 2-A).

Sensitivity analysis was performed to determine the source of heterogeneity, and the results showed that excluding one study [31] considerably reduced the between-study heterogeneity, resulting in a statistically significant increase in basal monocytic TLR4 density in persons with schizophrenia compared to HCs (Hedges' $g = 0.469$, 95% CI: 0.195 to 0.742, $p = 0.001$) (Fig 3-A). Heterogeneity analysis revealed significant heterogeneity within the AP-free subgroup ($Q = 12.74$, $df = 3$, $p = 0.005$, $\tau2 = 0.184$, $I^2 = 76.50\%$). The overall heterogeneity across all studies remained significant ($Q = 12.86$, $df = 4$, $p = 0.012$, $\tau2 = 0.127$, $I^2 = 68.91\%$), indicating moderate heterogeneity. Notably, the MFI is a relative unit that can vary significantly across laboratories [51]. The Galbraith plot confirmed the resilience of the included studies to outliers, confirming the absence of data points above the expected range (Supporting Information 2; Figure S1-A). The evaluation of publication bias ($z = -1.22$, $p = 1.462$) suggested no significant asymmetry in the funnel plot (Supporting Information 2; Figure S2-A).

### 3.5.  Basal percentage of TLR4-expressing monocytes: individual findings and quantitative synthesis

Five studies (518 individuals; 276 persons with schizophrenia and 242 HCs) assessed the basal percentage of TLR4+ monocytes [28,30,31,33]. Three studies investigated the outcome in AP-medicated persons with schizophrenia [30,33,36], one in individuals free of APs [31], and one in both settings [28]. Among AP-free participants [28,31], while one study found an increased percentage of TLR4+ monocytes in persons with schizophrenia compared to HCs [28], the other indicated no significant difference [31]. Among AP-medicated persons with schizophrenia [28,30,33,36], two studies found no significant difference between persons with schizophrenia and HCs [28,30], one study reported a lower percentage [33], and one reported a greater proportion of TLR4+ monocytes in persons with schizophrenia than in HCs [36].

The AP-free subgroup had a combined effect size of 0.399 (95% CI: -0.909 to 1.707), indicating a nonsignificant increase in TLR4+ monocytes in patients with schizophrenia. For the AP-medicated subgroup, the effect size was 0.158 (95% CI: -0.371 to 0.686), suggesting no significant difference. The overall analysis indicated a pooled effect size of 0.235 (95% CI: -0.245 to 0.715), reflecting a generally nonsignificant increase in TLR4+ monocytes in patients with schizophrenia compared to HCs. The test of group differences did not reveal a significant effect ($Q_b$ (1) $= 0.11$, $p = 0.737$), suggesting no significant difference between the AP-free and AP-medicated subgroups (Fig 2-B).

According to the sensitivity analysis, excluding individual studies did not significantly alter the overall effect size, indicating the robustness of the findings to the exclusion of individual studies (Fig 3-B). Heterogeneity analysis revealed significant heterogeneity in both the AP-free ($Q = 14.81$, $df = 1$, $p < 0.001$, $\tau2 = 0.83$, $I^2 = 93.25\%$) and AP-medicated subgroups ($Q = 24.67$, $df = 3$, $p < 0.001$, $\tau2 = 0.25$, $I^2 = 85.82\%$), as well as overall heterogeneity ($Q = 41.80$, $df = 5$,

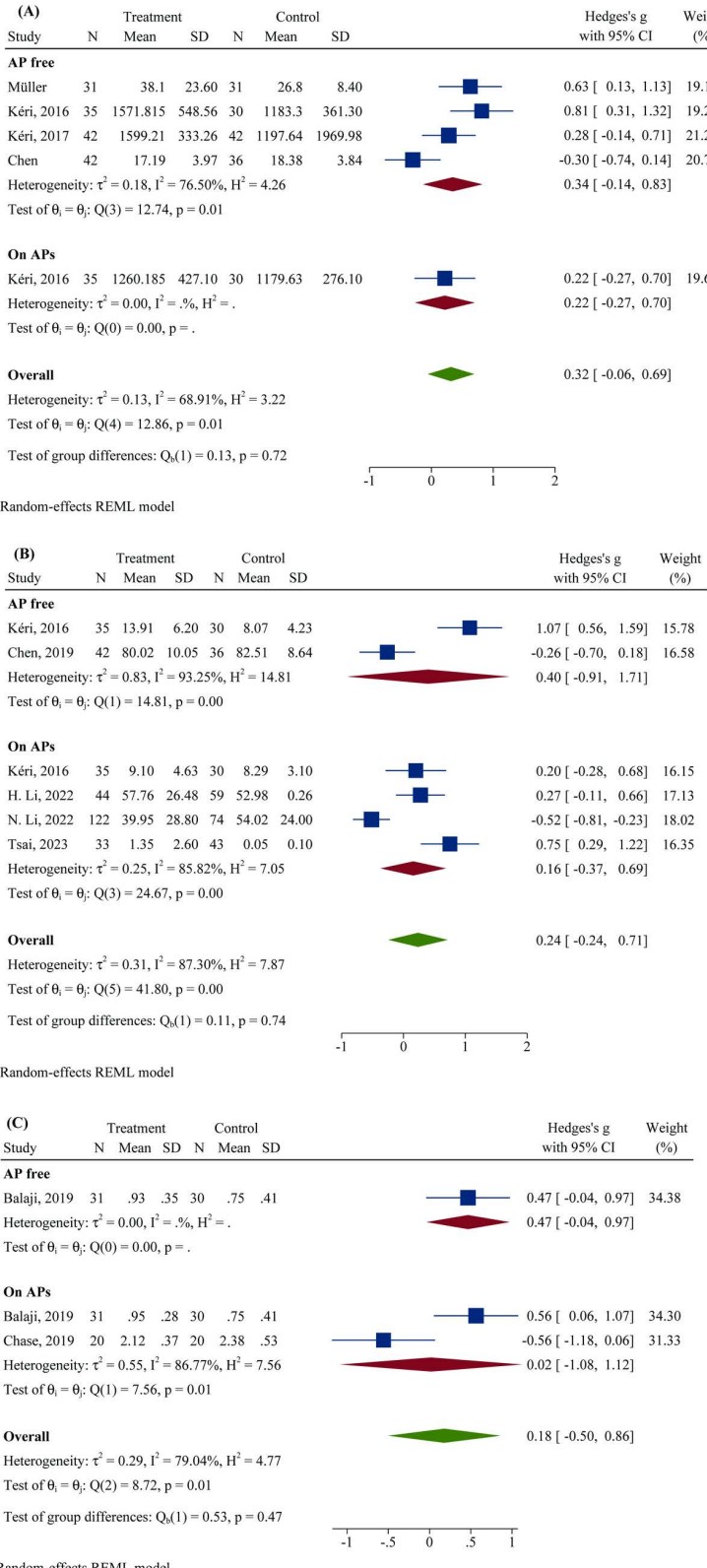

**Fig 2. Forest plot of Hedges' g, comparing the outcomes between persons with schizophrenia and healthy controls.** A) Basal surface monocytic TLR4 density, B) basal percentage of TLR4-expressing monocytes, and C) basal TLR4 gene expression in PBMCs (*Note: since the results for HCs were not reported after the 3-month follow-up in the Balaji et al. study, we assumed that the TLR mRNA expression for HCs after three months was roughly equivalent to their baseline expression).

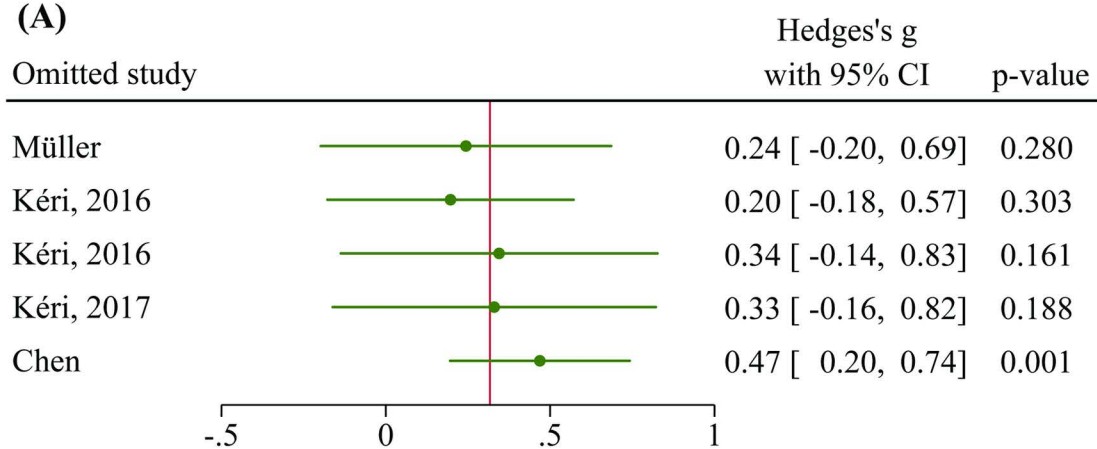

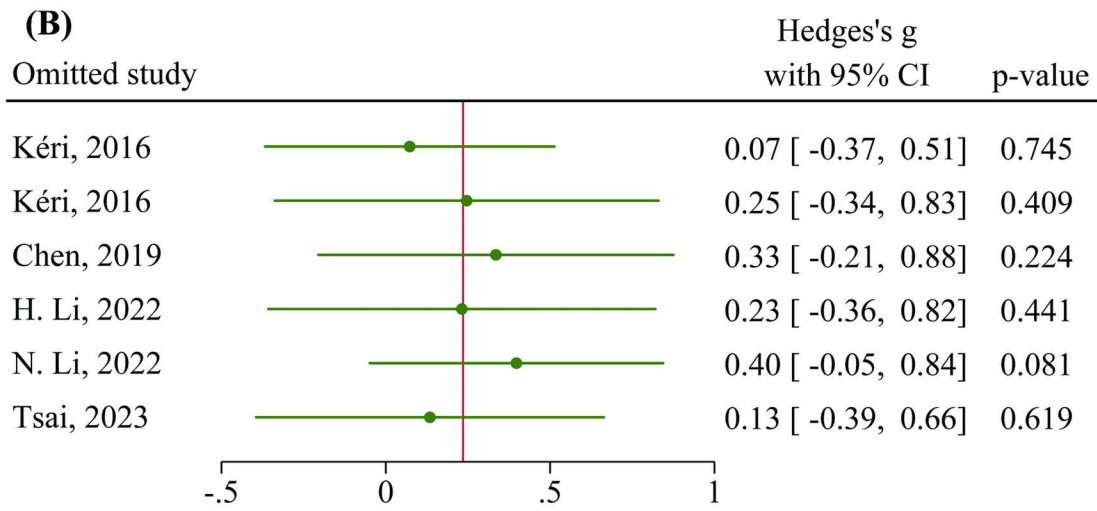

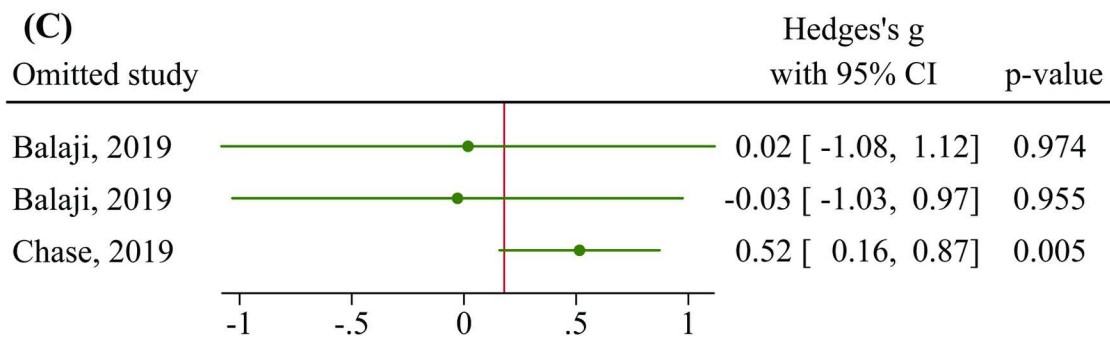

**Fig 3. Sensitivity analysis.** A) basal surface monocytic TLR4 density, B) basal percentage of TLR4-expressing monocytes, and C) basal TLR4 gene expression in PBMCs.

$p < 0.001$, $\tau 2 = 0.31$, $I^2 = 87.30\%$). Galbraith figures showed that all studies fell within the 95% CI of standardized Hedges' g, indicating no significant deviation from the overall trend. This alignment within the CI suggested no extreme outliers, ensuring consistent contributions to the meta-analysis and reinforcing the findings' reliability (Supporting Information 2; Figure S1-B). The assessment for publication bias showed no statistical significance, despite slight asymmetry ($z = -1.88$, $p = 0.133$) (Supporting Information 2; Figure S2-B).

## 3.6. Basal TLR4 gene/mRNA expression in PBMCs: individual findings and quantitative synthesis

Three studies focused on TLR4 gene/mRNA expression in PBMCs [29,34,35], with two comparing outcomes between persons with schizophrenia on APs and HCs [34, 35] and one assessing both drug-naïve and AP-medicated individuals [29]. TLR4 gene expression was evaluated in two studies (101 individuals; 51 persons with schizophrenia and 50 HCs) [29,34]. One study revealed upregulated TLR4 gene expression in drug-naïve persons with schizophrenia compared to HCs [29]. In the same study, TLR4 gene expression remained unchanged after three months of AP therapy [29]. The second study, which focused on actively medicated participants, revealed no significant difference in TLR4 gene expression between these individuals and HCs [34]. TLR4 mRNA expression was examined in one study (56 individuals) [35], which suggested that TLR4 expression was downregulated in persons with schizophrenia compared with HCs [35].

Pooled analysis across the AP-free and AP-medicated subgroups showed nonsignificant results. The overall effect size for TLR4 gene expression (0.179, 95% CI: –0.502 to 0.861) revealed a nonsignificant increase in TLR4 expression in patients with schizophrenia, with no significant difference between the subgroups ($Q_b$ (1) = 0.53, $p = 0.467$) (Fig 2-C). The sensitivity analysis demonstrated a significant change in the overall effect size (Hedges' g = 0.515, 95% CI = 0.159 to 0.872, $p = 0.005$) after omitting one study [34], indicating greater gene expression in persons with schizophrenia than in HCs (Fig 3-C). The heterogeneity analysis indicated substantial heterogeneity (Q = 8.72, df = 2, $p = 0.013$, $\tau 2 = 0.29$, $I^2 = 79.04\%$). The Galbraith plot indicated that no individual study deviated significantly from the overall trend (Supporting Information 2; Figure S1-C). No funnel plot asymmetry was observed ($z = 0.00$, $p = 1.000$) (Supporting Information 2; Figure S2-C).

## 3.7. TLR4 expression after stimulation: individual findings

Four studies (439 individuals; 239 persons with schizophrenia and 200 HCs) evaluated the TLR4 expression pattern after stimulation [22,30,31,33] (Table 2). All studies employed 4 [22] to 5 [30,31,33] hours of LPS stimulation, with 1 μg/mL [22] and 100 ng/mL [30,31,33] concentrations. One study also used polyinosinic: polycytidylic acid sodium salt (polyI:C), which mimics a viral stimulus and is known to interact with TLR3 [22]. One study evaluated the MFI as the outcome of interest [22], two studies assessed the percentage of TLR4+ monocytes [30,33], and one study assessed both [31].

Two studies assessed TLR4 density on the monocyte membrane following stimulation, both involving AP-free individuals or those with a brief duration of AP use [22,31]. One study found no statistically significant difference in the *raw* MFI values after LPS stimulation [22]. However, they observed a smaller increase in TLR4 expression in persons with schizophrenia than in HCs after stimulation using the following formula: (unstimulated MFI/stimulated MFI) [22]. Another study reported lower *raw* MFI values in individuals with schizophrenia than in HCs following LPS stimulation, without a significant difference in the *reduction rate*, calculated using the formula: (unstimulated MFI-stimulated MFI)/unstimulated MFI [31].

The authors concluded that persons with schizophrenia exhibit *weakened* monocytic TLR4 activation, as indicated by a lower *stimulated* MFI in persons with schizophrenia, despite similar *basal* monocytic TLR4 density between the two groups [31].

Three studies assessed the percentage of TLR4+ monocytes after stimulation [30,31,33]. One study focused on AP-free participants [31], and two were conducted among individuals who had been on a stable AP regimen for six months or longer [30,33]. Chen *et al.* reported a lower percentage of TLR4+ monocytes after LPS challenge in individuals with schizophrenia than in HCs [31], a finding consistent with that of H. Li *et al.* [30], suggesting a blunted monocytic TLR4 response to LPS stimulation. However, they observed no significant difference in the *reduction rate* of TLR4+ monocytes [(unstimulated % – stimulated %)/unstimulated %], aligning with the findings of N. Li *et al.* [33].

## 4. Discussion

This systematic review included 11 studies (473 persons with schizophrenia and 416 HCs) investigating *basal* TLR4 expression and 4 studies (239 persons with schizophrenia and 200 HCs) investigating *stimulated* TLR4 expression in individuals with schizophrenia. Meta-analyses were performed for three outcomes, including basal monocytic TLR4 density (MFI), basal percentage of TLR4+ monocytes, and basal PBMC's TLR4 gene expression. Subgroup analyses were performed according to AP status. We observed an overall trend toward increased monocytic TLR4 density in persons with schizophrenia (regardless of AP status) compared to HCs, although the difference was not significant. Notably, after sensitivity analysis and excluding one study [31], this difference reached statistical significance. No significant difference was observed between the groups in terms of the percentage of TLR4+ monocytes, despite a general increase in persons with schizophrenia. Similar results were obtained for TLR4 gene expression. However, excluding one study [34] indicated significantly greater TLR4 gene expression in persons with schizophrenia than in HCs. Nevertheless, the limited number of studies prevents definitive conclusions on this matter. Studies attributed to TLR4 expression after stimulation suggested weakened monocytic activation in individuals with schizophrenia.

Several factors might contribute to the discrepancies between the original studies, including differences in the type or dosage of AP medication, therapy duration, and characteristics of study participants (e.g., disease duration and severity). These factors are plausible given their potential to influence an individual's immune and inflammatory status [41]. For instance, a recent systematic review and meta-analysis revealed a significant reduction in pro-inflammatory cytokines following risperidone treatment, an effect not observed with clozapine [52]. Subgroup analyses, comparing first-episode psychosis to chronic patients, demonstrated that disease duration also influenced the extent of cytokine alterations. Risperidone treatment significantly lowered levels of IL-6 and TNF-α in patients with chronic disease, while this effect was not evident in individuals experiencing their first episode of psychosis [52]. Future investigations should encompass a broad range of medications, along with multiple follow-up assessments for a longer duration.

### TLR4 expression in individuals with schizophrenia

The first study on TLR4 expression in schizophrenia was performed by Chang et al. in 2011, who examined the proportion of people with schizophrenia with monocytic TLR4 mRNA expression [32]. A later study by Müller *et al.* indicated an increased monocytic TLR4 density in persons with schizophrenia compared to HCs [22]. Later investigations yielded varying results, with some reporting nonsignificant results [31] and others reporting increased [21,28] monocytic TLR4 density in individuals with schizophrenia. The heightened expression of

TLR4 was suggested to be a compensatory response to functional deficits in monocytes [22,31]. Subsequent investigations on the proportion of TLR4+ monocytes or TLR4 gene expression in persons with schizophrenia yielded inconclusive findings [28–31,33–35].

The TLR4/IL-1β/NF-κβ signaling pathway has been reported to play a role in various CNS conditions, such as cerebral ischemia and traumatic brain injury [53,54]. IL-1β and NF-κβ, which are downstream inflammatory mediators of the TLR4 pathway, are known to have notable effects on the brain by modulating neuroinflammation, neurogenesis, and neuro-degeneration [7,55,56]. Thus, Li *et al.* aimed to explore the TLR4/IL-1β/NF-κβ pathway in schizophrenia [30]. The authors reported a greater monocytic expression of IL-1β in persons with schizophrenia than in HCs in an unstimulated state. Additionally, there were nonsignificant tendencies toward elevated NF-κB and TLR4 in individuals with schizophrenia [30]. According to the authors, two possible mechanisms could explain the activation of the TLR4 pathway. First, the "leaky gut" hypothesis of schizophrenia suggests that gram-negative bacteria can enter the blood due to increased intestinal permeability, resulting in TLR4 signaling pathway activation [57,58]. Second, prenatal infections, including viral, bacterial, and proto-zoan infections, can induce maternal immune activation, resulting in nitrosative or oxidative stress responses [59]. Consistently, TLR4-induced mild inflammation has been observed in both first-episode [31] and chronic schizophrenia [33] patients.

## TLR4 and clinical and neuropsychological variables in individuals with schizophrenia

Disparities exist regarding the relationship between TLR4 and the clinical or cognitive features of people with schizophrenia. According to the study by Balaji *et al.*, TLR4 gene expression was not significantly correlated with positive or negative symptoms (assessed using the SAPS and SANS), either at baseline or after AP treatment [29]. Similarly, Li *et al.* did not detect any significant correlation between TLR4 levels and PANSS-positive, negative, general, or total scores [30]. This finding was in line with that of Chase *et al.,* who found no association between TLR4 mRNA levels and PANSS-positive scores in persons with schizophrenia [34]. In another study, which included 61 individuals with tardive dyskinesia (TD) and 61 individuals without TD (NTD), no associations were found between the TLR4 signaling pathway and PANSS scores [33].

Cognitive impairments and executive dysfunctions are integral parts of schizophrenia [60]. The presence of mild chronic inflammation may play a significant role in the development of these cognitive impairments [61]. Notably, increased TLR4 expression, myeloid differen-tiation primary response gene 88 (MyD88), and NF-κβ, which are components of the TLR4 signaling pathway, were detected in the prefrontal cortex of people with schizophrenia [14]. Dysfunctions in the prefrontal cortex result in executive dysfunctions and attention deficits, which are frequently observed in individuals with schizophrenia [62]. In a recent study, Li *et al.* indicated that TLR4 levels and cognitive function (assessed by the MATRICS Consensus Cognitive Battery [MCCB]) were negatively correlated at baseline in the TD group [33]. After LPS stimulation, a positive correlation was demonstrated. This may suggest that chronic subtle inflammation mediated by *basal* TLR4 can diminish cognitive function, while a rapid immune response mediated by *stimulated* TLR4 might benefit cognitive function in people with schizophrenia [33]. Additionally, according to the study by Keri *et al.*, in drug-naïve per-sons with schizophrenia, a higher percentage of TLR4+ monocytes was associated with poorer cognitive function, as indicated by a negative correlation with the Repeatable Battery for the Assessment of Neuropsychological Status (RBANS) test score. The RBANS assesses immedi-ate memory, language, visuospatial functions, attention, and delayed memory [63]. However,

these correlations did not reach statistical significance after eight weeks of AP treatment [28]. It was inferred that the association between inflammation and cognition might vary between individuals who have been receiving AP treatment and those with recent-onset schizophrenia [28]. In contrast, according to the Chen *et al.* study, increased basal monocytic TLR4 expression has a positive impact on visual learning and working memory in first-episode schizophrenia patients [31]. The reason for this inconsistency is unclear, but it might be linked to the physiological roles of TLRs in neuroplasticity [31].

### TLR4 and neuroimaging markers

To date, few studies have addressed the relationship between neuroimaging markers and TLR4 expression in patients with schizophrenia. According to the study by Li *et al.*, no regional or whole-brain cortical thickness was correlated with TLR4 levels in individuals with schizophrenia or in the control group [30]. Accordingly, the authors suggested that the impacts of TLR4 on cognition might occur through effects on the white matter rather than cortical thickness or gray matter volume [30]. However, future studies are required to shed light on neuroimaging alterations resulting from the effects of TLR4 on white and gray matter.

### Antipsychotic medications and TLR4 in schizophrenia

The relationship between immune responses and APs seems to be multifaceted. Notably, commonly prescribed APs possess anti-inflammatory properties, potentially reducing microglial activation [64,65]. Moreover, novel therapeutic options have been assessed, mostly for their anti-inflammatory effects [66]. APs affect various immunological functions, such as the number of T and B cells, T-cell differentiation, cytokine and chemokine secretion, and gene expression in PBMCs [28,67–70]. Only a few studies have sought to address the gap in understanding the impact of APs on TLR4 expression in persons with schizophrenia [28]. According to Keri *et al.,* while there was an increase in TLR4 before treatment in persons with schizophrenia compared to HCs, the difference was not significant after treatment with risperidone or olanzapine [28]. The authors suggested that APs might normalize TLR4 expression in persons with schizophrenia. In contrast, Balaji *et al.* reported no significant difference in TLR4 gene expression before and after three months of AP therapy in persons with schizophrenia, indicating that medication use did not influence TLR4 gene expression [29].

### Practical implications

This study revealed a trend towards elevated monocytic TLR4 density alongside diminished monocytic TLR4 activation in individuals with schizophrenia, indicating a dysregulated immune response that may play a crucial role in the pathophysiology of the disorder. These findings could open avenues for novel therapeutic strategies. Modulating TLR4 activity or targeting related downstream pathways through adjunctive therapies alongside standard APs holds promise for improving treatment outcomes. This is particularly relevant considering the potential link between TLR4 signaling and cognitive deficits, a critical and often challenging aspect of schizophrenia treatment.

### Limitations

This study has several limitations. First, there are few articles on this topic to date, making it difficult to establish a definitive relationship between TLR4 and schizophrenia. Second, the sample sizes in most of the studies were relatively small; hence, the findings might not be generalizable. Third, variations in methodologies and the assessment of diverse outcomes across

studies make it challenging to conduct a meta-analysis, given that the number of studies for each outcome was less than five. Fourth, only one study differentiated CD14$^+$ monocytes into classical (CD14$^+$ and CD16$^-$) and intermediate (CD14$^+$ and C16$^+$) subsets [36]. Finally, the high heterogeneity observed in several meta-analyses ($I^2$ > 75%), particularly within subgroups stratified by AP status poses challenges in clearly interpreting the results [71]. This variability likely arises from differences in study populations, methodologies, and treatment protocols. To address this, we employed a random-effects model with REML estimation to account for between-study variability, conducted subgroup analyses based on AP status to minimize confounding effects, and performed sensitivity analyses, which demonstrated the stability of our findings despite the heterogeneity. Additionally, publication bias assessments confirmed the robustness of our results.

## 5. Conclusions and further directions

The results of meta-analyses suggested a trend toward higher basal monocytic TLR4 density in persons with schizophrenia than in HCs. The differences in the basal percentage of TLR4$^+$ monocytes and basal TLR4 gene expression were not significant. According to qualitative data synthesis, persons with schizophrenia exhibit blunted monocytic TLR4 activation after stimulation. However, many unexplored questions should be answered in future studies; these include understanding the impact of AP usage and its specific type on TLR4 in people with schizophrenia and exploring the correlation between TLR4 and diverse clinical, neuropsychological, and neuroimaging characteristics of those with schizophrenia. Additionally, identifying the differences between acute and chronic activation of TLR4 in schizophrenia patients is crucial for obtaining a comprehensive understanding of this disease. Future studies should also account for potential confounding factors affecting the association between TLR4 and schizophrenia, such as TLR4 gene polymorphisms, AP medication type, duration and dosage, duration of disease, and different phases of schizophrenia (prodromal, active, residual). Additionally, although intermediate monocytes (CD14$^+$ and CD16$^+$) constitute a minor population (nearly 10%) of human monocytes [72], it would be fruitful for future research to consider monocyte heterogeneity.

## Supporting information

**Supporting Information 1. PRISMA Checklist.**
(DOCX)

**Supporting Information 2. Tables S1–S7 and Figures S1–S2 are compiled in this single PDF**.
(PDF)

**Supporting Information 3. The data supporting the findings is provided in this supplemental Excel file**.
(XLSX)

## Author contributions

**Conceptualization:** Melika Jameie, Sanaz Bordbar, Reza Samiee, Mobina Amanollahi, Mir Sajjad Aleyasin, Mohammad Reza Abdol Homayuni, Mehrdad Mozafar, Seyed Behnamedin Jameie, Shahin Akhondzadeh.

**Data curation:** Melika Jameie, Sanaz Bordbar, Reza Samiee, Mobina Amanollahi, Mir Sajjad Aleyasin, Mohammad Reza Abdol Homayuni, Mehrdad Mozafar.

**Formal analysis:** Mehdi Azizmohammad Looha.

**Methodology:** Melika Jameie, Sanaz Bordbar.

**Supervision:** Melika Jameie, Seyed Behnamedin Jameie, Shahin Akhondzadeh.

**Writing – original draft:** Melika Jameie, Sanaz Bordbar, Reza Samiee, Mobina Amanollahi, Mehdi Azizmohammad Looha, Mir Sajjad Aleyasin, Mohammad Reza Abdol Homayuni, Mehrdad Mozafar.

**Writing – review & editing:** Melika Jameie, Seyed Behnamedin Jameie, Shahin Akhondzadeh.

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
