## [Decision Letter · Decision Letter 0]

5 Dec 2024

PONE-D-24-33258Monocytic TLR4 expression and activation in schizophrenia: a systematic review and meta-analysisPLOS ONE

Dear Dr. Akhondzadeh,

Thank you for submitting your manuscript to PLOS ONE. After careful consideration, we feel that it has merit but does not fully meet PLOS ONE’s publication criteria as it currently stands. Therefore, we invite you to submit a revised version of the manuscript that addresses the points raised during the review process.

We look forward to receiving your revised manuscript.

Kind regards,

Ahmad Salimi

Academic Editor

PLOS ONE

**Journal Requirements:**

Please ensure that your manuscript meets PLOS ONE's style requirements, including those for file naming. The PLOS ONE style templates can be found at https://journals.plos.org/plosone/s/file?id=wjVg/PLOSOne_formatting_sample_main_body.pdf and https://journals.plos.org/plosone/s/file?id=ba62/PLOSOne_formatting_sample_title_authors_affiliations.pdf 2. In the online submission form, you indicated that The data underlying the results presented in the study are available from the corresponding author. Contact information: sakhond@yahoo.com All PLOS journals now require all data underlying the findings described in their manuscript to be freely available to other researchers, either a. In a public repository, b. Within the manuscript itself, or c. Uploaded as supplementary information.This policy applies to all data except where public deposition would breach compliance with the protocol approved by your research ethics board. If your data cannot be made publicly available for ethical or legal reasons (e.g., public availability would compromise patient privacy), please explain your reasons on resubmission and your exemption request will be escalated for approval.  3. PLOS requires an ORCID iD for the corresponding author in Editorial Manager on papers submitted after December 6th, 2016. Please ensure that you have an ORCID iD and that it is validated in Editorial Manager. To do this, go to ‘Update my Information’ (in the upper left-hand corner of the main menu), and click on the Fetch/Validate link next to the ORCID field. This will take you to the ORCID site and allow you to create a new iD or authenticate a pre-existing iD in Editorial Manager. 4. Your ethics statement should only appear in the Methods section of your manuscript. If your ethics statement is written in any section besides the Methods, please delete it from any other section. 5. As required by our policy on Data Availability, please ensure your manuscript or supplementary information includes the following:  A numbered table of all studies identified in the literature search, including those that were excluded from the analyses.   For every excluded study, the table should list the reason(s) for exclusion.   If any of the included studies are unpublished, include a link (URL) to the primary source or detailed information about how the content can be accessed.  A table of all data extracted from the primary research sources for the systematic review and/or meta-analysis. The table must include the following information for each study:  Name of data extractors and date of data extraction  Confirmation that the study was eligible to be included in the review.   All data extracted from each study for the reported systematic review and/or meta-analysis that would be needed to replicate your analyses.  If data or supporting information were obtained from another source (e.g. correspondence with the author of the original research article), please provide the source of data and dates on which the data/information were obtained by your research group.  If applicable for your analysis, a table showing the completed risk of bias and quality/certainty assessments for each study or outcome.  Please ensure this is provided for each domain or parameter assessed. For example, if you used the Cochrane risk-of-bias tool for randomized trials, provide answers to each of the signalling questions for each study. If you used GRADE to assess certainty of evidence, provide judgements about each of the quality of evidence factor. This should be provided for each outcome.   An explanation of how missing data were handled.  This information can be included in the main text, supplementary information, or relevant data repository. Please note that providing these underlying data is a requirement for publication in this journal, and if these data are not provided your manuscript might be rejected.   6. Please review your reference list to ensure that it is complete and correct. If you have cited papers that have been retracted, please include the rationale for doing so in the manuscript text, or remove these references and replace them with relevant current references. Any changes to the reference list should be mentioned in the rebuttal letter that accompanies your revised manuscript. If you need to cite a retracted article, indicate the article’s retracted status in the References list and also include a citation and full reference for the retraction notice.

Reviewers' comments:

Reviewer's Responses to Questions

**Comments to the Author**

1. Is the manuscript technically sound, and do the data support the conclusions?

Reviewer #1: Yes

Reviewer #2: Yes

2. Has the statistical analysis been performed appropriately and rigorously? 

Reviewer #1: Yes

Reviewer #2: Yes

3. Have the authors made all data underlying the findings in their manuscript fully available?

Reviewer #1: Yes

Reviewer #2: No

4. Is the manuscript presented in an intelligible fashion and written in standard English?

Reviewer #1: Yes

Reviewer #2: Yes

5. Review Comments to the Author

**Reviewer #1:**  Manuscript number: PONE-D-24-33258

Manuscript title: Monocytic TLR4 expression and activation in schizophrenia: a systematic review and meta-analysis

This is an interesting topic and an area to expand further research. While I believe the work presents an interesting topic, there are multiple issues with the content of the paper.

• Search strategy

• Please set up the method and describe how to select references and establish the logic or focus of this review. It must be comprehensive or logical in the focused topic as much as you can.

• Consider adding a summarizing table(s) and/or figures presenting some of the discussed mechanism and the flowchart of the study selection process.

**Reviewer #2: ** This study addresses an important topic and has a suitable overall structure. However, it requires more clarity and detail in certain sections:

1. Introduction

1. Please provide additional details about the selection of TLR4 as a specific receptor and its role in the pathophysiology of schizophrenia. Additionally, explain why this receptor is more significant compared to other immune pathways or molecules.

2. Elaborate further on the potential factors that might explain inconsistencies in previous study findings (e.g., differences in experimental methods, population characteristics, or medication effects).

3. Clearly specify how the findings of this systematic review can contribute to improving current therapeutic approaches or inform novel treatment strategies for schizophrenia.

2. Methods

1. Please provide more details about the use of keyword combinations (e.g., Boolean logic such as AND/OR) and any filters applied during the search process.

2. Clarify the number of studies retrieved at each stage of the search process and provide transparent reasons for their exclusion.

3. Explain why certain studies were categorized as “inconsistent with the research question.”

3. Results

1. Please explain how the high heterogeneity of the studies (I² > 75%) impacts the interpretation of the results.

4. Discussion

1. Highlight more explicitly how the findings of this study relate to practical applications or the development of new treatments for schizophrenia.

2. Provide further explanations about the potential factors contributing to inconsistencies in previous studies, including the effects of antipsychotic drugs, demographic differences, or variations in experimental methods.

Suggestion: Addressing these points will significantly enhance the quality and impact of the manuscript, establishing it as a reliable and valuable resource in this field.

6. PLOS authors have the option to publish the peer review history of their article (what does this mean? ). If published, this will include your full peer review and any attached files.

**Do you want your identity to be public for this peer review?** For information about this choice, including consent withdrawal, please see our Privacy Policy .

Reviewer #1: No

Reviewer #2: No

---

## [Author Response · Author response to Decision Letter 1]

10 Jan 2025

PONE-D-24-33258

Monocytic TLR4 expression and activation in schizophrenia: a systematic review and meta-analysis

PLOS ONE

Dear Dr. Akhondzadeh,

Thank you for submitting your manuscript to PLOS ONE. After careful consideration, we feel that it has merit but does not fully meet PLOS ONE’s publication criteria as it currently stands. Therefore, we invite you to submit a revised version of the manuscript that addresses the points raised during the review process.

We look forward to receiving your revised manuscript.

Kind regards,

Ahmad Salimi

Academic Editor

PLOS ONE

- Reply: Dear Dr. Ahmad Salimi,

Thank you very much for providing us with the opportunity to strengthen our research. We sincerely appreciate all the precious comments from you and the respected reviewers. Having carefully considered the comments and suggestions, we have made all the relevant changes to our manuscript as outlined below in an itemized, point-by-point manner. We sincerely hope that these changes meet the approval criteria of the esteemed reviewers and the editorial board.

Best Regards,

• Shahin Akhondzadeh; Pharm. D., PhD

Full Professor of Neuroscience; Psychiatry and Psychology Research Center, Roozbeh Hospital, Tehran University of Medical Sciences, Tehran, Iran; sakhond@yahoo.com

• Seyed Behnamedin Jameie; PhD

Full Professor of Anatomy and Neuroscience; Neuroscience Research Center and Department of Anatomy, Iran University of Medical Sciences, Tehran, Iran; jameie.sb@iums.ac.ir; behjame@gmail.com

Journal Requirements:

1. When submitting your revision, we need you to address these additional requirements. Please ensure that your manuscript meets PLOS ONE's style requirements, including those for file naming. The PLOS ONE style templates can be found at https://journals.plos.org/plosone/s/file?id=wjVg/PLOSOne_formatting_sample_main_body.pdf and https://journals.plos.org/plosone/s/file?id=ba62/PLOSOne_formatting_sample_title_authors_affiliations.pdf

o Reply: Thank you for your time and effort. The manuscript meets PLOS ONE's style requirements.

2. In the online submission form, you indicated that The data underlying the results presented in the study are available from the corresponding author. Contact information: sakhond@yahoo.com. All PLOS journals now require all data underlying the findings described in their manuscript to be freely available to other researchers, either a. In a public repository, b. Within the manuscript itself, or c. Uploaded as supplementary information. This policy applies to all data except where public deposition would breach compliance with the protocol approved by your research ethics board. If your data cannot be made publicly available for ethical or legal reasons (e.g., public availability would compromise patient privacy), please explain your reasons on resubmission and your exemption request will be escalated for approval.

o Reply: Thank you for mentioning this issue. In the revised manuscript, the data underlying the findings is uploaded as supplementary information. Accordingly, we revised the Data Availability Statement.

o Reply: Thank you very much. We will make sure that the ORDID IDs for corresponding authors are provided in Editorial Manager. However, here, we provide their ORCIDs.

Melika Jameie (First author): 0000-0002-2028-9935

Seyed Behnamedin Jameie (Corresponding author): 0000-0003-2062-4155

Shahin Akhondzadeh (Corresponding author): 0000-0002-2277-5101

o Reply: We removed the Ethics approval and consent to participate section at the end of the manuscript. In the revised version, the ethics statement only appears in the Methods section.

5. As required by our policy on Data Availability, please ensure your manuscript or supplementary information includes the following:

- A numbered table of all studies identified in the literature search, including those that were excluded from the analyses. For every excluded study, the table should list the reason(s) for exclusion.

o Reply: Thank you very much for your constructive feedback. According to the journal’s policy on data availability, we added this information in a new supplemental table (Supplementary Table S6). This is mentioned in the text: “Table S6 presents a numbered list of the 125 studies identified in the literature search that proceeded to full-text screening, along with the reasons for exclusion for each excluded study.” It should be noted that there might be more than one reason for exclusion for each excluded study.

o Reply: There are no included studies that are unpublished.

- A table of all data extracted from the primary research sources for the systematic review and/or meta-analysis. The table must include the following information for each study: Name of data extractors and date of data extraction, and Confirmation that the study was eligible to be included in the review.

o Reply: The Name of the data extractors and date of data extraction were added for the included studies in Supplementary Table S6. The confirmation that the study was eligible to be included is reported in Supplementary Table S6.

o Reply: All data extracted from each study for the reported systematic review and meta-analysis that would be needed to replicate our analyses can be found in Tables 1 and 2 of the manuscript. Additionally, we provided the data underlying the meta-analysis results as Supplementary information.

o Reply: All data is obtained from the main original articles. No other source has been used.

o Reply: Table S7 provides detail information regarding the Risk of bias evaluation for the outcome of interest for each study, using the Joanna Briggs Institute (JBI) Critical Appraisal tool.

- An explanation of how missing data were handled. This information can be included in the main text, supplementary information, or relevant data repository. Please note that providing these underlying data is a requirement for publication in this journal, and if these data are not provided your manuscript might be rejected.

o Reply: All information required for the meta-analyses was extracted from the main text or the main figures of the original manuscripts. For studies where the numbers were not reported in the main text, we included the following note in the table legends: “As the study results were shown in the figures (not tables), the numbers were extracted from the figures using https://automeris.io/WebPlotDigitizer/. Therefore, the extracted numbers might be slightly different from the exact numbers found by the authors.” Notably, the underlying data is provided as supplementary information in the revised version of our manuscript.

o Reply: Thank you again for all your time and effort. The reference list is complete and correct.

Reviewer 1:

This is an interesting topic and an area to expand further research. While I believe the work presents an interesting topic, there are multiple issues with the content of the paper.

1. Search strategy

2. Please set up the method and describe how to select references and establish the logic or focus of this review. It must be comprehensive or logical in the focused topic as much as you can.

3. Consider adding a summarizing table(s) and/or figures presenting some of the discussed mechanism and the flowchart of the study selection process.

o Reply: Dear reviewer,

Thank you very much for your kind words and for providing us with the opportunity to strengthen our research. We sincerely appreciate your precious comments. Having carefully considered the comments and suggestions, we have made all the relevant changes to our manuscript as outlined below in an itemized, point-by-point manner. The revisions requested have been track-changed and highlighted within the manuscript in response.

1. Reply: Thank you very much for your precious comment. Our main research question was to investigate the basal expression and activation pattern of TLR4 in PBMCs, including monocytes, among individuals with schizophrenia compared to HCs. We conducted a systematic search in three databases, including MEDLINE (via PubMed), Scopus, and Web of Science. Then, we used the “Patient, Intervention, Comparison, and Outcome” (PICO) model to formulate our research question [1]:

- Patient: Persons with schizophrenia

- Intervention/exposure: not applicable

- Comparison: HCs

- Outcome(s):

1. TLR4 basal expression patterns in PBMCs

2. TLR4 activation patterns in PBMCs

Then, two PICO elements were used to convert our research question into a search strategy:

a. Patient-related MeSH Terms and text word terms: (“schizophrenia”, “schizophrenic”, “schizophrenias”, “psychosis”, and “psychotic”) AND

b. Outcome-related MeSH terms and text word terms: “toll-like”, “toll-like receptor”, “toll-like receptor 4”, and “TLR”

All the information is written in the “Methods section > Objectives and review questions/ Study design, information sources, and search strategy”. However, since we taught that providing the whole search strategy within the manuscript text might be space-consuming, we provided the detailed search strategy in Supplemental Tables S3-S5. Sincerely, please let us know if any other clarifications are required considering the search strategy.

MEDLINE (via PubMed) No.

1. (Toll-like receptors [MeSH Terms] OR “toll-like receptors” [Title/Abstract] OR TLR [Title/Abstract] OR “toll-like receptor” [Title/Abstract] OR “Toll-like” [Title/Abstract] OR “Toll-Like receptor 4” [Title/Abstract] OR TLR4 [Title/Abstract] OR TLR-4 [Title/Abstract] OR “Toll-Like receptor 4” [MeSH Terms]) 73579

2. (Schizophrenia [MeSH Terms] OR schizophrenia [Title/Abstract] OR psychotic [Title/Abstract] OR psychosis [Title/Abstract] OR psychotic disorders [MeSH Terms] OR “psychotic disorders” [Title/Abstract] OR “psychotic disorder” [Title/Abstract] OR schizophrenias [Title/Abstract] OR schizophrenic [Title/Abstract]) 228052

#1 AND #2 112

Web of Science

1. (TS= (“Toll-like receptors”) OR TS= (“toll-like receptor”) OR TS= (TLR) OR TS= (“Toll-like”) OR TS= (“Toll-Like receptor 4”) OR TS= (TLR4) OR TS= (TLR-4)) 87616

2. (TS= (Schizophrenia) OR TS= (psychotic) OR TS= (psychosis) OR TS= (“psychotic disorders”) OR TS= (“psychotic disorder”) OR TS= (schizophrenias) OR TS= (schizophrenic)) 266798

#1 AND #2 171

Scopus

1. (INDEXTERMS("Toll-like receptors") OR TITLE-ABS("toll-like receptors") OR TITLE-ABS(TLR) OR TITLE-ABS("toll-like receptor") OR TITLE-ABS(Toll-like) OR TITLE-ABS("Toll-Like receptor 4") OR TITLE-ABS(TLR4) OR TITLE-ABS(TLR-4) OR INDEXTERMS("Toll-Like receptor 4")) 111233

2. (TITLE-ABS-KEY (Schizophrenia ) OR TITLE-ABS-KEY (psychotic ) OR TITLE-ABS-KEY (psychosis ) OR TITLE-ABS-KEY ("psychotic disorders" ) OR TITLE-ABS-KEY ("psychotic disorder" ) OR TITLE-ABS-KEY (schizophrenias ) OR TITLE-ABS-KEY (schizophrenic )) 340059

#1 AND #2 339

2. Reply: Thank you very much for your thoughtful feedback. The references were selected according to the instructions below.

- First, a total of 622 records from the three databases were initially imported into EndNote software (version X9, Clarivate Plc). Then, duplicates were removed, yielding 452 records for initial screening. Then, two independent researchers (R.S. and S.B.) screened all the 452 records using titles and abstracts. Among these articles, 125 potentially eligible articles remained for full-text screening by three independent authors (M.J., S.B., and R.S.), including one expert author (M.J.). Notably, the references of these articles were also manually searched for any potentially related article that was not retrieved initially by database searching (Hand searching method). The records that did not align with the predefined eligibility criteria were excluded. In cases of discrepancies, conflicts were resolved by reaching a consensus among all authors (including the two corresponding authors).

- Among these 125 articles, 114 did not align with the predefined eligibility criteria, leaving 11 original articles aligning with the research question. The reason for the exclusion of each study is reported in the revised version of the manuscript as Supplementary Table S6. The information regarding the selection of studies is reported in the “Methods section > Selection process”, as well as the “Results section > Literature search and study selection”. According to your valuable comment, to ensure integrity, completeness, and logic behind selecting the references, the table below also shows the reason(s) for excluding each single study. Please let us know of there are any other clarifications required concerning selecting the studies.

Table S6. Studies identified in the literature search that proceeded to full-text screening

Title First author Eligibility Exclusion reason/ Extractors

1. JNK signaling mediates aspects of maternal immune activation: importance of maternal genotype in relation to schizophrenia risk R.L Openshow [2]

Exclude Animal study

2. CNS Macrophages and Infant Infections A. Oschwald [3]

Exclude Review

3. Role of inflammation in epilepsy and neurobehavioral comorbidities: Implication for therapy Y.N. Paudel [4]

Exclude Review

4. The role of the gut microbiota in the pathophysiology of mental and neurological disorders M.M. Pusceddu [5]

Exclude Review

5. N-3 polyunsaturated fatty acids and clozapine abrogates poly I: C-induced immune alterations in primary hippocampal neurons B.M.M. Ribeiro [6]

Exclude In-vitro

6. Lack of Helios During Neural Development Induces Adult Schizophrenia-Like Behaviors Associated With Aberrant Levels of the TRIF-Recruiter Protein WDFY1 A. Sancho-Balsells [7]

Exclude Animal study

7. Targeting the NLRP3 Inflammasome-Related Pathways via Tianeptine Treatment-Suppressed Microglia Polarization to the M1 Phenotype in Lipopolysaccharide-Stimulated Cultures J. Slusarczyk [8]

Exclude In-vitro

8. Importance of the immune system in mediating plasticity of t

---

## [Decision Letter · Decision Letter 1]

29 Jan 2025

Monocytic TLR4 expression and activation in schizophrenia: a systematic review and meta-analysis

PONE-D-24-33258R1

Dear Dr. Shahin Akhondzadeh,

We’re pleased to inform you that your manuscript has been judged scientifically suitable for publication and will be formally accepted for publication once it meets all outstanding technical requirements.

Kind regards,

Ahmad Salimi

Academic Editor

PLOS ONE

Additional Editor Comments (optional):

Reviewers' comments:

Reviewer's Responses to Questions

**Comments to the Author**

1. If the authors have adequately addressed your comments raised in a previous round of review and you feel that this manuscript is now acceptable for publication, you may indicate that here to bypass the “Comments to the Author” section, enter your conflict of interest statement in the “Confidential to Editor” section, and submit your "Accept" recommendation.

Reviewer #1: All comments have been addressed

Reviewer #2: All comments have been addressed

2. Is the manuscript technically sound, and do the data support the conclusions?

Reviewer #1: Yes

Reviewer #2: Yes

3. Has the statistical analysis been performed appropriately and rigorously? 

Reviewer #1: Yes

Reviewer #2: Yes

4. Have the authors made all data underlying the findings in their manuscript fully available?

Reviewer #1: Yes

Reviewer #2: Yes

5. Is the manuscript presented in an intelligible fashion and written in standard English?

Reviewer #1: Yes

Reviewer #2: Yes

6. Review Comments to the Author

Reviewer #1: (No Response)

Reviewer #2: (No Response)

7. PLOS authors have the option to publish the peer review history of their article (what does this mean? ). If published, this will include your full peer review and any attached files.

**Do you want your identity to be public for this peer review?** For information about this choice, including consent withdrawal, please see our Privacy Policy .

Reviewer #1: **Yes: ** Masoomeh Dadkhah

Reviewer #2: No

---

## [Editor Report · Acceptance letter]

PONE-D-24-33258R1

PLOS ONE

Dear Dr. Akhondzadeh,

I'm pleased to inform you that your manuscript has been deemed suitable for publication in PLOS ONE. Congratulations! Your manuscript is now being handed over to our production team.

Kind regards,

on behalf of

Dr. Ahmad Salimi

Academic Editor

PLOS ONE